# Establishing pregnancy-specific haematological reference intervals in Ghana; a three-center cross-sectional study

**Nicodemus Larbi Simpong[1], Charity Tenu Afefa[2], Leander Yimpuri[2], Betty Akum[2], Afia Safo[2], Simon-Junior Edziah[2], David Larbi Simpong[2], Patrick Adu**  [2]*

1 Maternity Ward, 37 Military Hospital, Accra, Ghana, 2 Department of Medical Laboratory Science, School of Allied Health Science, University of Cape Coast, Cape Coast, Ghana

* Patrick.adu@ucc.edu.gh

## Abstract

### Background

Abnormal intra-pregnancy haematological variables are associated with adverse feto-maternal outcomes. However, the reference intervals (RIs) employed in sub-Saharan Africa to inform clinical decisions are generally imported. Since RIs are influenced by age, geographical location, and race, we hypothesized that context specific RIs should be established in Ghana to contextualize intra-pregnancy decision making.

### Methods

This cross-sectional study retrospectively retrieved data of 333 pregnant women with no known clinically determined intra-pregnancy complications; 22 participants in their first trimester (T1; 1–13 weeks), 177 in their T2 (14–27 weeks), and 132 in T3 (28–41 weeks). RIs for haematological parameters were non-parametrically determined at 2.5th and 97.5th percentiles in accordance with CLSI guidance document EP28-A3c. Two-sample comparisons were undertaken using Wilcoxon rank-sum tests whereas more than two-sample comparisons were undertaken using Kruskal-Wallis test. Statistical significance was set at p <0.05 under the two-tailed assumptions.

### Results

In accordance with WHO trimester-specific haemoglobin cutoffs, anaemia prevalence was a moderate (T1: 36.4%; 8/22 & T2: 31.6%; 56/177) to severe (T3:68.0%; 90/132) public health problem. Additionally, 9.3% (31/333) individuals had high gestational haemoglobin levels (Hb >13.0 g/dL). Moreover, haemoglobin (T2: 8.6–14.3 vs T3: 7.5–13.6 g/dL), MCH (T2: 22.5–69.8 vs T3: 21.6–31.9 pg), MCHC (T2: 30.2–51.8 g/L vs T3: 30.5–37.9 g/L), TWBC (T2: 4.0–13.4 vs T3: 4.1–13.0 x $10^9$/L) required trimester specific RIs, compared to RBC (2.8–5.1 x $10^{12}$/L), MCV (66.2–100.2 fL), and platelet counts (106.3–388.3 x $10^9$/L) that each required combined reference intervals.

**Data Availability Statement:** All relevant data are within the manuscript and its Supporting Information files.

**Funding:** The author(s) received no specific funding for this work.

**Competing interests:** The authors have declared that no competing interests exist.

**Abbreviations:** RBC, Red blood cell count; HCT, haematocrit; MCV, mean cell volume; MCH, mean cell haemoglobin; MCHC, mean cell haemoglobin concentration; RDW-SD, red cell distribution width–standard deviation; TWBC, total white blood cell count; PLT, platelet count; CLSI, Clinical and Laboratory Standards Institute.

## Conclusions

The intra-pregnancy haematological RIs determined have appreciable lower limits; there is the need to determine context-specific thresholds for haematological variables predictive of positive and/or adverse maternal and infant health outcomes.

## Introduction

Haematological testing is routinely requested during the gestational period to assist in the monitoring and management of pregnancy. To assist in the accuracy of test interpretation, haematological reference intervals (RIs) are important considerations in clinical decision-making during pregnancy. Since RIs are influenced by age, gender, dietary habits, and race, the Clinical and Laboratory Standard Institute (CLSI) recommends population-specific RIs to situate and contextualize clinical decisions [1]. Previous research in the sub-region have determined that most haematological RIs in sub-Saharan Africa have lower limits lesser than western-centric RIs [2–6]. For example, in the definition of anaemia in adults, the World Health Organisation (WHO) recommends that haemoglobin cutoff of 12.5 g/dL (females) and 13.5 g/dL (males) be used in Caucasian population compared to 11.5 g/dL (females) and 12.5 g/dL (males) for individuals from sub-Saharan Africa [7]. These different reference intervals have been established in view of the fact that race is an important consideration in reference intervals setting.

For anaemia classification during the gestational period, the WHO further recommends haemoglobin cutoffs of 11.0 g/dL, 10.5 g/dL, and 11.0 g/dL respectively for first, second and third trimesters among Caucasians [8]. However, haemoglobin cutoffs for gestational period in sub-Saharan Africa is not available. As different haemoglobin cutoffs have been established among non-pregnant adult Caucasian and sub-Saharan African women, it will be interesting to establish cutoffs haemoglobin as well as other haematological values for pregnant women in these different races. We are of the opinion that pregnancy specific RIs should be established within the sub-region to contextualize clinical decision-making process. If a value of 11.5 g/dL is considered to be the lower limit for haemoglobin among non-pregnant female in sub-Saharan Africans, then adopting the Caucasian-centric haemoglobin cutoff of 11.0 g/dL for first trimester would mean that the onset of pregnancy is associated with a 0.5 g/dL drop in the first trimester compared to a 1.5 g/dL drop in Caucasian women (12.5 g/dL in non-pregnant Caucasian vs 11.0 g/dL pregnant Caucasian women). As the onset of pregnancy is associated with physiological haemo-dilution [9, 10], there is the need to establish lower limit haemoglobin values for pregnant women in sub-Saharan Africa. Thus, setting the trimester 1 haemoglobin threshold at 11.0 g/dL for sub-Saharan African women (non-pregnant haemoglobin threshold at 11.5 g/dL) may be inappropriate and would likely lead to misclassification and misdiagnosis. This study therefore retrospectively examined the data from pregnant women with no clinically diagnosed complications and used the non-parametric criteria to set the reference intervals at the 2.5th and 97.5th percentile for red cells and its indices, white blood cells and platelets. Since the Clinical and Laboratory Standard Institute (CLSI) stipulates that partitioning of reference intervals should be evaluated [1], we also assessed whether separate trimester-specific or combined gestational reference intervals were required for any of the haematological variables. Since intra-pregnancy haematological reference intervals have not been explored within the sub-Saharan African region, we aimed to generate empirical baseline data to begin to address this knowledge gap, and stimulate discussion as well as further research that will ultimately lead to improved context-specific intra-pregnancy clinical decision-making process.

## Materials and methods

### Study participants

The population of interest in this study was pregnant women who had no known clinically diagnosed complications in pregnancy. Overall, data from a total of 333 pregnant women were included in the present study. The study was conducted between June, 2021 and September, 2021. The classification scheme used for staging pregnancy were: trimester 1 (1–13 weeks); trimester 2 (14–27 weeks); trimester 3 (28–41 weeks).

### Study design/setting

This cross-sectional study retrospectively retrieved data from 333 pregnant women aged between 14 to 47 years across Northern and the Southern zones in Ghana. Specifically, for the Southern zone, data was retrieved from the Antenatal centers at Tafo government hospital in Ashanti region, and Eastern Regional Hospital, Koforidua; for the Northern zone, data was retrieved from the Care Diagnostics Centre, Wa in the Upper West region.

### Sample size

In accordance with recommendations from the Clinical and Laboratory Standards Institute (CLSI), the minimum sample size required for determination of reference interval is 120 reference participants. Out of the 333 participants data included in the present study, 22, 179 and 132 participants were in trimesters 1, 2 and 3 respectively.

### Inclusion criteria

Data from all pregnant women who had visited the study sites for their antenatal care were included in the study. All participants were on the following medications: folic acid (5 mg daily), oral iron tablets (200 mg three times daily) and oral multivitamin tables; these medications are generally called routine pregnancy drugs within the Ghana Health Services as they are part of the routine medical care for all pregnant women.

### Exclusion criteria

Data from pregnant women who were suffering from any underlying condition(s) such as diabetes, hypertension, eclampsia, pre-eclampsia, diagnosed sickle cell anaemia, HELLP syndrome or malaria, HIV or hepatitis B virus infection were excluded from the study. Additionally, pregnant women with underlying clinically diagnosed liver and/or kidney disease were also excluded from the study.

### Ethical consideration

All the study protocols were reviewed and approved (ethical clearance ID: UCCIRB/CHAS/ 2019/82) by the Institutional Review Board of the University of Cape Coast. Additionally, study was approved by the respective Institutional Research Boards of the various hospitals before data acquisition commenced. No personal identifiers such as name, phone numbers etc. were included as part of the data collection to ensure that the data cannot be directly traced to any participant. This study did not require informed consent from the respective participants since the data was retrieved retrospectively and de-identified to prevent any possibility of tracing data to any participant.

### Experimental protocols

**Full blood count.**   The full blood count parameters recorded in the present study for each participant was undertaken using a 5-part Sysmex 20N haematology analyser (Sysmex Corp., Japan) being used in each facility. Per the prevailing protocols in each of the participating laboratory, control samples (low, normal and high) are run daily as quality control measure/check on analyser performance prior to assaying patient sample. Only when the respective haematology analyser passes the quality control tests is the analyser used to assay patient samples. Additionally, each of these daily quality control sample data (low, normal and high) are cumulatively evaluated monthly as a quality assurance measure to identify trends in analyser performance that may require immediate action. Anaemia was classified based on the trimester-specific cut-offs set by the World Health Organization; trimester 1 (Hb <11.0 g/dl), trimester 2 (Hb <10.5 g/dl), and trimester 3 (Hb <11.0 g/dl). Maternal haemoglobin and adverse birth outcomes have been shown to follow a U-shaped curve where both low and high gestational haemoglobin are associated with adverse pregnancy outcomes such as preterm, gestational diabetes, stillbirth, pre-eclampsia, and small-for-gestational age [11–13]. Therefore, in accordance with previous meta-analyses report [14], intra-pregnancy Hb >13.0 g/dL was classified as high maternal haemoglobin to classify participants with high gestational haemoglobin.

**Sickling slide test.**   All the sickling status recorded in the present study for each participant was determined using the 2% sodium metabisulphite test in accordance with previous protocols [15]. As per the protocols of the laboratories of each hospital, positive and negative controls were set up each day to control their respective sickling reagent.

**Glucose 6 phosphate dehydrogenase test (G6PD).**   The G6PD status of each participant recorded in the study was determined using the methaemoglobin reductase test [15] by the respective hospitals. All testing was internally controlled using both positive and negative controls per each patient sample test.

**Data management and data analyses.**   Data was double-entered and checked in a Microsoft Excel 2016 Spreadsheet. Data entry were at each study site was undertaken independently by two researchers after which they were validated by the research supervisor as a quality control process to ensure that data was accurately captured. For statistical analysis, the Statistical Package for the Social Sciences (SPSS) version 22 (IBM Inc., USA), GraphPad Prism software version 8.0.2 (GraphPad Prism Inc., San Diego, USA) and Stata IC version 16.0 (Stata Corp, College Park, TX, USA) were utilized. Normality of data was assessed using the D'Agostino-Pearson omnibus test prior to analyses. Categorical variables were summarized as percentages. The reference intervals (RIs) were non-parametrically determined at 2.5th and 97.5th percentiles as recommended by CLSI. The Mann-Witney U test was performed to determine statistical differences between two groups, whereas more than two groups comparison was undertaken using the Kruskal-Wallis test. All statistical significance testing were undertaken using the two-tailed assumptions; p-value of <0.05 was considered statistically significant. The reference intervals were determined using the non-parametric approach; the 2.5th percentile of the distribution of haematological test results was used as the lower limit, while the 97.5th percentile was used as the upper limit [16] with associated 90% confidence intervals for each limit. The decision as to whether a combined reference intervals or separate inter-trimester reference interval was to be adopted was made based on the standard deviations ratio >1.5 in accordance with recommendations by Harris and Boyd [17].

## Results

The socio-demographic details of the participants are presented in Table 1. Participants were fairly distributed across the three study sites. However, whereas 87.3% were within the 20–

**Table 1. Socio-demographic details of participants.**

| Variable | N (%) |
|---|---|
| **Region** | |
| Ashanti | 110 (33.0) |
| Eastern | 115 (34.5) |
| Upper West | 108 (32.4) |
| Total | 333 |
| **Age (years)** | |
| 14–19 | 28 (8.4) |
| 20–29 | 152 (45.6) |
| 30–39 | 139 (41.7) |
| 40–47 | 14 (4.2) |
| Total | 333 |
| **Gestational age** | |
| Trimester 1 | 22 (6.6) |
| Trimester 2 | 179 (53.9) |
| Trimester 3 | 132 (39.6) |
| Total | 333 |

39-year bracket, 8.4% were teenagers. Also, whereas 53.9% of participants were in the second trimester, only 6.6% of the participants were in their first trimester.

The immuno-haematological variables of participants are presented in Table 2. Blood group O (45.3%) was the predominant ABO blood type with blood group AB being the least represented (4.8%). In terms of Rh D typing, overwhelming majority (87.4%) typed as Rh D positive. Also, whereas only 4.8% of participants had defective qualitative G6PD deficiency, 7.5% tested positive for the sickling slide test. Furthermore, whereas 46.5% of participants had low haemoglobin level and therefore anaemic, 9.3% had higher haemoglobin levels. When participants haemoglobin levels were considered per trimester, 36.4% (95% CI: 0.1719–0.5934), 31.6% (95% CI: 0.2487–0.3904) and 68.2% (95% CI: 0.5951–0.7601) respectively of participants in trimester 1, 2 and 3 were anaemic.

Participants' haemoglobin levels were compared with the demographic data (Fig 1). Overall, the median haemoglobin significantly differed across the trimesters of pregnancy. The median haemoglobin was higher in trimester 1 (11.80 g/dl) but decreased progressively across trimesters 2 (11.00 g/dl) and 3 (10.50 g/dl) [Fig 1A: p = 0.0232 (trimester 1 vs trimester 3); p = 0.0005 (trimester 2 vs trimester 3)]. There was a general trend toward higher median haemoglobin levels with advancing age of participants as the respective median haemoglobin levels for 14–19 years, 20–29 years, 30–39 years, and 40–49 years were 10.35 g/dl, 10.80 g/dl, 11.00 g/dl, and 11.10 g/dl [Fig 1B; p = 0.049 (14–19 years vs 30–39 years group; p = 0.0278 (14–19 years vs 40–49 years group)]; a similar trend was also observed when data was explored for trimester 2 (Fig 1C) and trimester 3 (Fig 1D) although the differences did not reach statistical significance. Also, the median haemoglobin level was significantly higher in participants from Ashanti region (11.80 g/dl) compared to median haemoglobin level of participants from Eastern region (10.50 g/dL) or Upper Est region (10.30 g/dl) [Fig 1E: p <0.0001 (Ashanti vs Eastern region); p <0.0001 (Ashanti vs Upper West region)]. Similarly, the median haemoglobin levels in trimester 2 was significantly higher in participants from Ashanti region (11.80 g/dl) compared to participants from Eastern (10.80 g/dl) or Upper West (10.30 g/dl) regions (Fig 1F: p = 0.0031 vs Eastern region; p <0.0001 vs Upper West region). Moreover, the median haemoglobin in trimester 3 was higher in participants from

**Table 2. Immuno-haematological variables of study participants.**

| Variables | N (%) |
|---|---|
| **ABO blood type** | |
| A | 54 (16.2) |
| AB | 16 (4.8) |
| B | 86 (25.8) |
| O | 151 (45.3) |
| Missing | 26 (7.3) |
| Total | 333 |
| **Rh D type** | |
| D positive | 291 (87.4) |
| D negative | 16 (4.8) |
| Missing | 26 (7.8) |
| Total | 333 |
| **G6PD status** | |
| Full defect | 7 (2.1) |
| Partial defect | 9 (2.7) |
| No defect | 317 (95.2) |
| Total | 333 |
| **Sickling test** | |
| Positive | 25 (7.5) |
| Negative | 284 (85.3) |
| Missing | 24 (7.2) |
| Total | 333 |
| **Haemoglobin classification** | |
| Low | 154 (46.5) |
| Normal | 146 (44.1) |
| High | 31 (9.3) |
| Total | 333 |
| **Anaemia prevalence per trimester** | |
| Trimester 1 | 8/22 (36.4) |
| Trimester 2 | 56/177 (31.6) |
| Trimester 3 | 90/132 (68.2) |
| Total | 331 |

Anaemia was classified based on the trimester-specific cut-offs set by the World Health Organization; trimester 1 (Hb <11.0 g/dl), trimester 2 (Hb <10.5 g/dl), and trimester 3 (Hb <11.0 g/dl); in accordance with previous meta-analyses report [14], intra-pregnancy Hb >13.0 g/dL was classified as high maternal haemoglobin.

Ashanti region (11.65 g/dl) compared to the Upper West (10.50 g/dl) region (Fig 1G: p = 0.0240).

The haematological variables were explored (Table 3) as to whether a combined reference range or trimester-based reference range was warranted in accordance with recommendations from CLSI [1]. Since only 22 participants were in trimester 1, trimester 1 was excluded from this section of the analysis. Although there were statistically significant differences in MCV (p = 0.0004), MCH (p <0.0001), MCHC (p <0.0001), RDW-SD (p = 0.0001) and platelet counts (p = 0.026) between trimester 2 and 3 (Table 3), not all of these comparative parameters had the ratio of the respective standard deviations greater than the 1.5 threshold as recommended by Harris and Boyd [17]. Overall, when the standard deviations of haematological

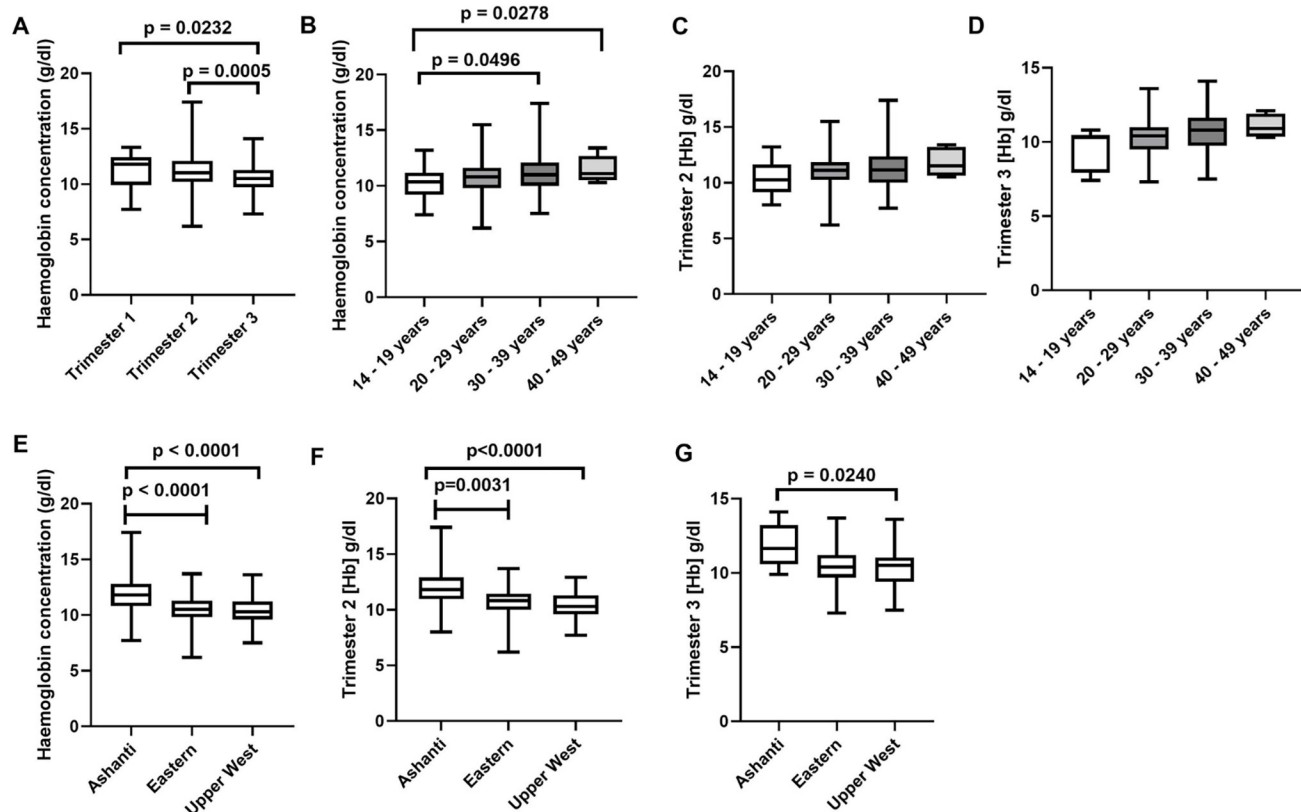

**Fig 1. Haemoglobin levels of participants stratified per socio-demographic variables.** Fig A compares the median haemoglobin levels of participants across the trimesters. Fig B compares the haemoglobin levels of all participants per age categories. Fig C compares the haemoglobin levels of participants in trimester 2 per age categories. Fig D compares the haemoglobin levels of participants in trimester 3 per age categories. Fig E compares the haemoglobin levels of all per the region. Fig F compares the haemoglobin of participants in trimester 2 per study site. Fig G compares the haemoglobin levels of participants in trimester 3 per study site. [All statistical comparisons were undertaken using the Kruskal-Wallis test with Dunn's multiple comparison correction test; All boxplots were constructed with whiskers indicating minimum and maximum data points].

parameters in trimester 2 and trimester 3 were compared, HCT, MCH, RDW-SD, TWBC met the required standard deviation threshold (standard deviations ratio >1.5) to warrant a separate reference range per each of trimester 2 and 3.

**Table 3. Comparison of the median and respective standard deviations of haematological variables of study participants.**

| Variable | Trimester 2 | | Trimester 3 | | p-value | SD2/SD1 |
|---|---|---|---|---|---|---|
| | Median | SD | Median | SD | | |
| RBC x $10^{12}$/L | 3.9 | 0.7 | 3.8 | 0.5 | 0.8479 | 1.4 |
| HCT (%) | 33.1 | 33.3 | 31.3 | 3.9 | **0.0002** | **8.5** |
| MCV (fL) | 85.9 | 8.2 | 82.7 | 8.1 | **0.0004** | 1.0 |
| MCH (pg) | 28.9 | 33.3 | 26.9 | 2.7 | **<0.0001** | **12.3** |
| MCHC (g/L) | 33.5 | 31.1 | 33.1 | 16.6 | **<0.0001** | **2.0** |
| RDW-SD | 42.1 | 31.4 | 40.6 | 4.9 | **0.0001** | **6.4** |
| TWBC x $10^9$/L | 7.3 | 6.3 | 7.4 | 1.9 | 0.2923 | **3.3** |
| PLT x $10^9$/L | 225.0 | 66.1 | 199.0 | 62.7 | **0.0026** | 1.1 |

RBC: Red blood cell count; HCT: haematocrit; MCV: mean cell volume; MCH: mean cell haemoglobin; MCHC: mean cell haemoglobin concentration; RDW-SD: red cell distribution width–standard deviation; TWBC: total white blood cell count; PLT: platelet count.

**Table 4. Trimester-specific and combined haematology reference intervals of study participants.**

| Variable | Variables with separate reference ranges | | | | | | | |
|---|---|---|---|---|---|---|---|---|
| | T2 | | | | T3 | | | |
| | N | Median | Ref. values | 90% CI | N | Median | Ref. values | 90% CI |
| Hb (g/dl) | 177 | 11.0 | 8.6–14.3 | 6.2–8.9; 13.5–17.4 | 132 | 10.5 | 7.5–13.6 | 7.3–8.4; 13.0–14.1 |
| HCT (%) | 177 | 33.1 | 26.0–49.6 | 18.7–27.1; 42.3 – | 132 | 31.3 | 24.5–41.4 | 23.6–25.7; 39.1–42.3 |
| MCH (pg) | 177 | 28.9 | 22.5–69.8 | 20.6–23.9; 36.5 – | 132 | 26.9 | 21.6–31.9 | 17.1–22.5; 31.2–33.2 |
| MCHC (g/L) | 177 | 33.5 | 30.2–51.8 | 27.4–31.0; 37.8 – | 132 | 33.1 | 30.5–37.9 | 30.2–30.6; 36.4 – |
| RDW-SD | 170 | 42.1 | 35.0–55.9 | 33.9–35.6; 53.0 – | 132 | 40.6 | 33.2–51.9 | 15.2–34.3; 49.1–55.5 |
| TWBC (x10$^9$/L) | 177 | 7.3 | 4.0–13.4 | 3.7–4.5; 12.6 – | 132 | 7.4 | 4.1–13.0 | 3.6–4.8; 11.4–14.4 |
| Variable | Variables with combined reference ranges | | | | | | | |
| | N | Median | Ref. values | 90% CI | | | | |
| RBC (x10$^{12}$/L) | 309 | 3.9 | 2.8–5.1 | 2.0–2.9; 4.9–5.4 | | | | |
| MCV (fL) | 309 | 85.9 | 66.2–100.2 | 65.6–68.4; 97.6–100.6 | | | | |
| PLT (x10$^9$/L) | 309 | 225.0 | 106.3–388.3 | 60.0–125.0; 363–408 | | | | |

RBC: Red blood cell count; HCT: haematocrit; MCV: mean cell volume; MCH: mean cell haemoglobin; MCHC: mean cell haemoglobin concentration; RDW-SD: red cell distribution width–standard deviation; TWBC: total white blood cell count; PLT: platelet count. Note: the upper limit of the 90% CI of HCT, MCH, MCHC, TWBC and RDW for T2 and MCHC for T3 are not indicated as they were very large to be considered technically feasible. [All reference intervals were calculated using the non-parametric assumptions].

Table 4 gives the reference ranges and their respective 90% CI determined for various haematological parameters. The haematological parameters that required separate reference intervals included haemoglobin [T2 (8.6–14. 3 g/dL) vs T3 (7.5–13.6 g/dL)], HCT [T2 (26.0–49.6%) vs T3 (24.5–41.4%)], MCH (22.5–69.8 pg) vs T3 (21.6–31.9 pg)], MCHC [T2 (30.2–51.8 g/L) vs (30.5–37.9 g/L)], RDW-SD [T2 (35.0–55.9) vs T3 (33.2–51.9)], TWBC [T2 (4.0–13.4 x10$^9$/L) vs T3 (4.1–13.0 x10$^9$/L)]. Additionally, the haematological parameters that required combined intra-pregnancy reference intervals included RBC (2.8–5.1 x 10$^{12}$/L), MCV (66.2–100.2 fL) and platelet count (106.3–388.3 x 10$^9$/L).

## Discussion

Reference intervals are critical in decision making process in the clinical settings. Traditionally, reference intervals used in most clinical laboratories in sub-Saharan African countries are imported from the countries/regions where the respective analysers were procured. The literature however, strongly argues that socio-demographic and geographic variations affect various haemato-biochemical reference ranges [2, 18]. Although the WHO has provided haemoglobin cut-off per trimester of pregnancy to guide decision making [8], we tested the hypothesis as to whether context-specific intra-pregnancy reference intervals for haematological parameters was required. This is important considering the variation in the haemoglobin cut-off level among non-pregnant females adult Caucasians compared to sub-Saharan Africans (12.5 vs 11.5 g/dL) [7]. As pre-pregnancy haemoglobin cut-off values in sub-Saharan African women are 1 g/dL lower than their Caucasians counterparts, we argued that a similar variation is possible among pregnant population from these two races. Adopting the same trimester-specific haemoglobin threshold values for pregnant women from different geographical location may not be realistic and therefore necessitates empirical research to interrogate the status quo.

Using the WHO recommended haemoglobin cut-off per trimester [8], our study estimated 46.5% overall anaemia prevalence among pregnant women in Ghana which is indicative of severe public health concern. Furthermore, when the data was explored per trimester, the

anaemia prevalence was 36.4%, 31.6% and 68.2% respectively for trimester 1, 2 and 3. As much as we agree that this might be indicative of a moderate-to-severe public health problem calling for public health campaign and action, we are of the opinion that this may not represent the true state of intra-pregnancy anaemia in Ghana. The supposed anaemia prevalence is exaggerated simply because the threshold used might not have been derived directly from the sub-Saharan context. Our trimester-specific reference intervals derived for haemoglobin levels [trimester 2 (8.6–14.3 g/dL) and trimester 3 (7.5–13.6 g/dL)] provides a provocative starting point to guide future research in this area. There is an urgent need for longitudinal studies that tracks a cohort of women prior to pregnancy and then through pregnancy to bring clarity as to the intra-pregnancy reference intervals that must be applicable within the sub-Saharan context. Compared with Caucasians, the haemoglobin cut-offs level for sub-Saharan Africans non-pregnant females is 1 g/dL lower (12.5 g/dL vs 11.5 g/dL). In Caucasians pregnant women, anaemia is defined as haemoglobin levels less than 11.0 g/dL, 10.5 g/dL, 11.0 g/dL for 1st, 2nd and 3rd trimesters respectively [8]. The classification of anaemia among sub-Saharan African pregnant women is not available. Since a difference of 1 g/dL haemoglobin value was observed between Caucasian and sub-Saharan African non-pregnant women, by inference we are tempted to assume a threshold of 10.0 g/dL, 9.5 g/dL and 10.0 g/dL for 1st, 2nd and 3rd trimester for sub-Saharan Africans pregnant women. If the present data is re-evaluated in light of the assumptions proposed, the anaemia prevalence among the study cohort reduces to 22.7% (5/ 22; trimester 1), 21.2% (19/177; trimester 2) and 34.1% (45/132; trimester 3). We are however not able to make any categorical inferences at this point since our study design does not permit such audacious recommendations. We therefore propose further research that employs longitudinal design that as well take into consideration birth outcomes to allow for exhaustive data exploration to derive context-specific haematological reference intervals that are directly applicable to pregnant women in the sub-Saharan Africa region. A recent meta-analysis posited that intra-pregnancy haemoglobin thresholds greater than 13.0 g/dL was associated with adverse birth outcomes just as lower haemoglobin levels [14]. Whether this proposed upper haemoglobin limit will be equally applicable in the sub-Saharan African context is an open question requiring urgent exploration. Therefore, this is a key area that requires further research in sub-Saharan Africa to determine haematological limits that are predictive of safe and better child and maternal outcomes to inform clinical decisions.

Previous research in the sub-region that sought to determine context-specific haemato-bio-chemical reference intervals have published ranges that differed from Western-centric values that accompany automated analysers. For example, using a cross-sectional study that recruited prospective blood donors, Addai-Mensah et al. [3] determined haematology reference intervals in non-pregnant females that had lower limit threshold than Horiba haematology analyser-supplied ranges; haemoglobin (10.22–15.50 g/dL vs 11.5–15.2 g/dL), MCV (68.4–92.0 fL vs 77.0–97.0 fL), and TWBC (3.28–7.85 x10$^9$/L vs 3.5–10.0 x10$^9$/L). Other similar studies in the sub-region and from elsewhere [3, 6, 19, 20] have additionally published similar haematological results to indicate that direct transference of these Western-centric reference ranges in clinical practice may lead to misleading diagnosis. Since all the research undertaken in sub-Saharan Africa points to the need to drop the lower limits (and in some cases the upper limits as well) of the reference intervals for haematological variables in the non-pregnant women, we are of the opinion that adopting the Western-centric intra-pregnancy haematology reference intervals will obfuscate prompt clinical decision making. The present study is proposing for the use of trimester-specific reference intervals for haemoglobin, haematocrit, mean cell haemoglobin (MCH), mean cell haemoglobin concentration (MCHC), red cell distribution width (RDW) and total white blood cell count compared to a combined reference intervals for other haematological variables such as red blood cell count (RBC), mean cell volume (MCV), and

platelet counts. Although our study found statistically significant differences in the median of most of the haematological parameters, RBC count, MCV, and platelet counts between trimester 2 and 3, did not passed the standard deviation ratio test [17]. As per the CLSI recommendations, statistically significant differences alone do not qualify for partitioning of reference ranges; decision to partition reference intervals between sub-groups should be informed by physiological relevance and/or standard deviation or z-test [1]. We are opining that adopting indigenous reference intervals as presented herein would lead to improved and context-specific therapeutic decisions. For example, Dosoo et al. determined that the platelet counts for non-pregnant females in Ghana should be appropriately set at 89–403 x $10^9$/L [6] with a lower limits comparatively lower than Caucasian-centric platelet count reference interval (150–400 x $10^9$/L) that accompanies haematological analysers. In a previous study among Danish pregnant women cohort, Milman et al. [21] determined platelet count reference intervals of 146–361 x $10^9$/L and 139–364 x $10^9$/L for trimesters 2 and 3 respectively. Comparatively, our proposed combined intra-pregnancy platelet count reference interval (106.3–388.3 x $10^9$/L) has a lower limit lesser than the interval proposed by Milman et al. study based on cohort of Danish pregnant women. The implications of these proposed ranges are that whereas intra-pregnancy platelet count of <150 x $10^9$/L would be considered thrombocytopenic (based on analyser-supplied reference intervals) and requiring monitoring as per the usually adopted Western-centric reference intervals, only intra-pregnancy platelet counts <106.3 x $10^9$/L will require such monitoring within the Ghanaian context. In the same vein, setting the trimester 2 and 3 haemoglobin reference intervals to 8.6–14.3 g/dL and 7.5–13.6 g/dL respectively would also mean that the classification scheme for intra-pregnancy anaemia determination ought to be updated accordingly to realistically reflect the sub-Saharan African context. What remains to be determined is the thresholds which could duly serve as the critical limits requiring emergency actions during pregnancy. For example, thrombocytopenia has traditionally been classified as mild (platelet count 100–149 x $10^9$/L), moderate (50–99 x $10^9$/L) and severe (<50 x$10^9$/L). Since the present study and other research [2, 3, 22] from the sub-region are advocating for dropping the lower limit for platelet counts to less than 110 x $10^9$/L, well-controlled longitudinal studies that will correlate pregnancy outcomes with intra- and post-partum platelet counts are warranted to inform context-specific policy formulation. Similarly, anaemia in women have been categorized into mild (10.0 -<11.5 g/dl), moderate (7.0 –<10.0 g/dL) and severe (<7.0 g/dl). Thus, further work is needed to be able to identify the corresponding cutoffs for trimester-specific mild, moderate and severe anaemia in order to contextualize clinical decisions and policy directions in the sub-region.

The sample size for trimester 1 was less than the required sample size of 120; this prevented us from determination of the trimester 1-specific reference intervals of the haematological variables. Moreover, the cross-sectional nature of the present study is such that we were unable to correlate our reference intervals with clinical outcomes as a means of validating our findings. This would have been an invaluable addition considering that a previous meta-analysis [14] has implicated haemoglobin values >13.0 g/dL to be associated with adverse pregnancy outcomes. Furthermore, in the absence of pregnancy outcome data, we were not able to explore the possibility of whether using continuous reference intervals adequately served the haematological changes during pregnancy. Since gestational period is characterized by profound physiological changes as a consequence of the foetal maturation process, discrete partitioning of the intra-pregnancy reference intervals might not adequately represent the complex variations in haematological variables. Therefore, future longitudinal study designs should consider simultaneously comparing discrete and continuous haematological reference intervals to ascertain which better reflect the dynamics in pregnancy. Recently, the CALIPER cohort study [23] concluded that continuous reference intervals better represented the complex changes in

haematological variables within the paediatric age range instead of the traditionally used discrete partitioning of these reference intervals. A similar data exploration would likely improve haematological test interpretation as well as improve intra-partum and post-partum decision making process. Our present study is limited by its cross-sectional nature which prevented us from acquiring data on pregnancy outcomes to enable such deeper data exploration and inferences. In spite of these acknowledged limitations, our study clearly demonstrate that the lower limits of all the intra-pregnancy haematology reference intervals are lesser than the corresponding Caucasian-centric ones. This is suggestive of a need to re-consider the haematological results threshold used in gestation period within the sub-Saharan African context to improve haematological test interpretation and management decisions.

## Supporting information

**S1 File. Data underlying the publication.**
(XLSX)

## Acknowledgments

We would like to thank the staff of the Laboratory Units of the Tafo Government hospital, Ashanti Region, Ghana, Eastern Regional hospital, Koforidua, Ghana, and the Care Diagnostic Centre, Wa, Upper West region for their assistance during our data collection. We are also grateful to MLS Evans Duah of Cape Coast Teaching hospital, Ghana, for assisting in aspects of data analyses.

## Author Contributions

**Conceptualization:** Nicodemus Larbi Simpong, Patrick Adu.

**Data curation:** Charity Tenu Afefa, Leander Yimpuri, Betty Akum, Afia Safo, Simon-Junior Edziah, David Larbi Simpong, Patrick Adu.

**Formal analysis:** David Larbi Simpong, Patrick Adu.

**Investigation:** Charity Tenu Afefa, Leander Yimpuri, Betty Akum, Afia Safo, Simon-Junior Edziah.

**Methodology:** Patrick Adu.

**Project administration:** David Larbi Simpong, Patrick Adu.

**Resources:** Charity Tenu Afefa, Leander Yimpuri, Betty Akum, Afia Safo, Simon-Junior Edziah, David Larbi Simpong, Patrick Adu.

**Supervision:** Nicodemus Larbi Simpong, Patrick Adu.

**Validation:** Nicodemus Larbi Simpong.

**Writing – original draft:** Patrick Adu.

**Writing – review & editing:** Nicodemus Larbi Simpong, Charity Tenu Afefa, Leander Yimpuri, Betty Akum, Afia Safo, Simon-Junior Edziah, David Larbi Simpong.

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
