## [Decision Letter · Decision Letter 0]

4 Apr 2022

PONE-D-22-04279Establishing pregnancy-specific haematological reference intervals in Ghana; a three-center cross-sectional study.PLOS ONE

Dear Dr. Adu,

Thank you for submitting your manuscript to PLOS ONE. After careful consideration, we feel that it has merit but does not fully meet PLOS ONE’s publication criteria as it currently stands. Therefore, we invite you to submit a revised version of the manuscript that addresses the points raised during the review process.

We look forward to receiving your revised manuscript.

Kind regards,

Orvalho Augusto, MD, MPH

Academic Editor

PLOS ONE

Journal Requirements:

Additional Editor Comments:

This is an important report to improve pregnancy outcomes. The authors make a very compelling argument for building hematologic reference intervals specific to African subpopulations, and in particular for pregnancy.

Issues:

1. Please, add line enumeration.

2. As both reviewers point out there is an important amount of typos and orthographic issues to fix.

3. The authors declared that the dataset is fully available. I did not see such data.

4. There are so many abbreviations. Please have a list of abbreviations at the end of the manuscript.

5. The authors removed participants with malaria, and many other important conditions. However, nothing is mentioned about HIV (interestingly hepatitis B is). Why?

6. In the sample size subsection:

- the last sentence starting at “The lower and upper reference…” should be removed. That is not part of sample size considerations.

7. The “Reference interval estimation” subsection should be merged with the “Data management and data analyses”

8. Data management and data analyses

- Please cite the software used

- For Excel, please add as well the version

9. Table 1:

- Please a row for overall total

10. Table 2:

- Please a row for overall total

- For hemoglobin classification, please add a footnote explaining the limits

11. Table 3:

- Add a footnote explaining the abbreviations

12. Figure 1:

- It is dangerous the mixture boxplots and line plots with presumably errors whiskers (which are in fact percentiles). Please separate these and explain in the footnote how the errors in the line plots are computed

13. Figure 2:

- There are no dotted lines. There are dashed lines.

Reviewers' comments:

Reviewer's Responses to Questions

**Comments to the Author**

1. Is the manuscript technically sound, and do the data support the conclusions?

Reviewer #1: Partly

Reviewer #2: Partly

2. Has the statistical analysis been performed appropriately and rigorously? 

Reviewer #1: Yes

Reviewer #2: Yes

3. Have the authors made all data underlying the findings in their manuscript fully available?

Reviewer #1: Yes

Reviewer #2: Yes

4. Is the manuscript presented in an intelligible fashion and written in standard English?

Reviewer #1: Yes

Reviewer #2: Yes

5. Review Comments to the Author

Reviewer #1: The authors sought to determine haematological reference intervals for different trimesters of pregnancy among pregnant women which can be used to make clinical decisions. Below are some suggestions to improve the quality of the manuscript.

Abstract

1. CLSI should be written out in full since it is not a standard abbreviation

2. Please clarify the version of the guidance document: is C28A2992 referring to C28-A3?

3. It is not clear why Hb >13 stated here and in the discussion should be considered as being high and linked to unfavorable pregnancy outcomes

4. It might be good to limit the number of decimal points for the various parameters as follows:

Hb - 1, MCHC - 1, RBC - 2, MCH - 1, MCV - 1 MCH - 1, WBC - 1, and PLT - 0 (in the manuscript)

5. Please indicate in the conclusion the comparison for which the determined reference intervals are lower

Keywords

6. the keywords look a bit too long and repetitive, and could be revised to:

Haematological; Reference intervals, Intra-pregnancy; Sub-Saharan Africa, pregnancy

Introduction

7. CLSI is Clinical and Laboratory Standards Institute, and not Clinical Laboratory Standards Institute

8. Please provide references for statements attributed to CLSI, and also references for previous research in Sub-Saharan Africa which authors indicate showed lower values

9. Please note NCCLS is an older name of CLSI, kindly update to CLSI

10. The last sentence of the introduction sounds like a conclusion and could be revised to explain the aim of the study

Methods

11. Please indicate the make and manufacturer of the haematology analyzers used for the analysis

12. Please indicate Quality Assurance/Quality Control measures put in place to ensure reliable results

13. Please give references for the sickling and G6PD tests

Results

14. Reference values are supposed to be determined using apparently healthy individuals. Inclusion of the Hepatitis B positive individuals will affect the determined values. Please exclude these individuals from the calculations

15. Presentation of the reference intervals in a tabular form will make it easier for the ready than the figures

16. The minimum of 120 individuals recommended by the CLSI for the determination of reference intervals is to make it possible to calculate 90% confidence intervals around the lower and upper values. The confidence intervals seem to be missing in the manuscript.

Discussion, References

17. The discussion is well written but could be updated with more references comparing similar studies.

18. the reference section looks too short. Kindly update

Reviewer #2: This manuscript by Adu et al. establishes reference intervals in a cohort of healthy pregnant women across trimesters for key haematology parameters. The authors are congratulated on their work as significant evidence gaps continue to exist in evidence-based cut-offs for test interpretation in pregnancy. However, further clarification and select changes are needed, particularly in the methods section. Specific comments are below:

Abstract:

- Suggest to remove "Deranged" and replace with "Abnormal"

- Suggest to include weeks of gestation for each trimester, space allowing

-

Methods:

- Exclusion criteria: Please clarify whether individuals with acute illness at the time of collection and/or use of prescribed medication (e.g. thyroid supplementation)

- Study participants: Please clarify whether iron supplementation was captured during assessment. In addition, was any longitudinal assessment evaluated at the end of the pregnancy?

- Reference Interval Determination: Please add confidence intervals for each estimate. CLSI guidelines recommends reporting 90% CIs

- Experimental Protocols: There is insufficient information in this section. For every assay, the manufacturer analyzers should be reported as well as their method (i.e. traceability) and performance throughout study duration (i.e. CV for QC levels). In addition, multiple centers were included in this study - were any center-specific differences in methodology evaluated?

- Data Analyses: CLSI guideline recommends using the Harris and Boyd method to test for partitions (i.e. gestational-age specific differences). Please clarify why MW and KW tests were selected. In addition, please clarify if outliers were removed and whether normality of data were assessed prior to statistical testing.

Results:

- Figure 1: Suggest to include these graphs for every parameter in supplemental methods.

- Table 1: Please clarify if individuals were excluded on the basis of positive HepB status.

- Table 3: Suggest to include RIs in this table as well (or an additional table) rather than in the graphs. In addition, was the differential evaluated?

Discussion:

- The discussion mainly focuses on Hgb levels and their classification based on WHO standards. It should be highlighted that the WHO standards may not suit the ethnic population of Ghana and it is potentially not a public health problem. If no pregnant individuals or their neonates had serious complications, should this be considered abnormal? Suggest to include a comment on this.

6. PLOS authors have the option to publish the peer review history of their article (what does this mean?). If published, this will include your full peer review and any attached files.

Reviewer #1: No

Reviewer #2: **Yes: **Mary Kathryn Bohn

---

## [Author Response · Author response to Decision Letter 0]

19 May 2022

Comment Response Page

EDITORIAL 

1. Please, add line enumeration Line numbers added 

2. As both reviewers point out there is an important amount of typos and orthographic issues to fix. The manuscript has been revised to address these 

3. The authors declared that the dataset is fully available. I did not see such data.

 The minimal data set underlying the results described in the manuscript has been uploaded as a Supporting Information file 

4. There are so many abbreviations. Please have a list of abbreviations at the end of the manuscript. List of abbreviations included Page 21;

Lines 147-155

5. The authors removed participants with malaria, and many other important conditions. However, nothing is mentioned about HIV (interestingly hepatitis B is). Why? Hepatitis B infected participants’ data have been excluded; data has subsequently been re-analysed. All data from HIV positive pregnant women were not included in the initial data collection. Page 6; Line 145

6. In the sample size subsection:

- the last sentence starting at “The lower and upper reference…” should be removed. That is not part of sample size considerations. “The lower and upper reference limits were determined non-parametrically using the 2.5th and 97.5th percentiles respectively” has been deleted from the “Sample size” section in the revised manuscript. Page 6; Line 131-136

7. The “Reference interval estimation” subsection should be merged with the “Data management and data analyses” The “reference interval estimation” section has been merged with the “Data management and data analyses section.” Pages 8-9; Line 174 - 190

8. Data management and data analyses

- Please cite the software used

- For Excel, please add as well the version The Data management and data analyses section of the manuscript has been revised to include “Data was double-entered and checked in a Microsoft Excel 2016 Spreadsheet. For statistical analysis, the Statistical Package for the Social Sciences (SPSS) version 22 (IBM Inc., USA), GraphPad Prism software version 8.0.2 (GraphPad Prism Inc., San Diego, USA) and Stata IC version 16.0 (Stata Corp, College Park, TX, USA) were utilized.” Page 8;

Line 174 – 178.

9. Table 1:

- Please a row for overall total Row totals have been calculated Page 9; line 197-198

10. Table 2:

- Please a row for overall total

- For hemoglobin classification, please add a footnote explaining the limits Row totals have been calculated

Manuscript has been revised to include a footnote

“Anaemia was classified based on the trimester-specific cut-offs set by the World Health Organization; trimester 1 (Hb <11.0 g/dl), trimester 2 (Hb <10.5 g/dl), and trimester 3 (Hb <11.0 g/dl); in accordance with previous meta-analyses report (4), intra-pregnancy Hb >13.0 g/dL was classified as high maternal haemoglobin.” Page 10; line 207-208

Line 209 - 211

11. Table 3:

- Add a footnote explaining the abbreviations Footnote defining abbreviations have been added. Page 12; line 249 - 251

12. Figure 1:

- It is dangerous the mixture boxplots and line plots with presumably errors whiskers (which are in fact percentiles). Please separate these and explain in the footnote how the errors in the line plots are computed

 All graphs have been converted to Boxplots; a footnote “(All box plots were constructed with whiskers indicating minimum and maximum data points)” has been added. Page 11; Line 228 – 231.

13. Figure 2:

- There are no dotted lines. There are dashed lines. Figure 2 has been converted into a table (table 4) as suggested by both reviewer 1 & 2 Page 13; line 254 - 268

REVIEWER 1 

Abstract 

1. CLSI should be written out in full since it is not a standard abbreviation Done Page 2; line 29

2. 2. Please clarify the version of the guidance document: is C28A2992 referring to C28-A3? Yes. Please see reference Page 2; line 38

3. 3. It is not clear why Hb >13 stated here and, in the discussion, should be considered as being high and linked to unfavorable pregnancy outcomes This has been clarified under section “full blood count”; this is from a meta-analyses report by Young et al (2019) “Young MF, Oaks BM, Tandon S, Martorell R, Dewey KG, Wendt AS. Maternal hemoglobin concentrations across pregnancy and maternal and child health: a systematic review and meta-analysis. Ann N Y Acad Sci. 2019;1450(1):47-68.” Page 7; Line 163-164

4. 4. It might be good to limit the number of decimal points for the various parameters as follows:

Hb - 1, MCHC - 1, RBC - 2, MCH - 1, MCV - 1 MCH - 1, WBC - 1, and PLT - 0 (in the manuscript). All decimals have been limited to 1 in all tables. 

5. Please indicate in the conclusion the comparison for which the determined reference intervals are lower This is applicable to all the parameters analysed in the current study. 

Keywords 

6. the keywords look a bit too long and repetitive, and could be revised to:

Haematological; Reference intervals, Intra-pregnancy; Sub-Saharan Africa, pregnancy Manuscript revised to reflect suggestions made by reviewer Page 3; line 52-53

Introduction 

7. CLSI is Clinical and Laboratory Standards Institute, and not Clinical Laboratory Standards Institute Manuscript revised to read Clinical and Laboratory Standard Institute (CLSI). Page 4; line 83

8. Please provide references for statements attributed to CLSI, and also references for previous research in Sub-Saharan Africa which authors indicate showed lower values Done. 

9. Please note NCCLS is an older name of CLSI, kindly update to CLSI Manuscript revised to uniformly read CLSI 

10. The last sentence of the introduction sounds like a conclusion and could be revised to explain the aim of the study. The last section of the introduction has been revised to read “Since intra-pregnancy haematological reference intervals have not been explored within the sub-Saharan African region, we aimed to generate empirical baseline data to begin to address this knowledge gap, and stimulate discussion as well as further research that will ultimately lead to improved context-specific intra-pregnancy clinical decision-making process.” Page 5; line 114 - 117

Methods 

11. Please indicate the make and manufacturer of the haematology analyzers used for the analysis The section has been revised accordingly to include “5-part Sysmex 20N haematology analyser (Sysmex Corp., Japan)” Page 7; line 160

12. Please indicate Quality Assurance/Quality Control measures put in place to ensure reliable results The “Data was double-entered and checked in a Microsoft Excel 2016 Spreadsheet” in the “Data management and data analyses” has been elaborated by adding “Data entry were at each study site was undertaken independently by two researchers after which they were validated by the research supervisor as a quality control process to ensure that data was accurately captured” to reflect QA procedures employed during data acquisition Page 8; line175 – 178.

13. Please give references for the sickling and G6PD tests Reference provided in the revised manuscript “Barabara J. Bain IB, Michael A Laffan, S. Mitchell Lewis. Dacie and Lewis Practical Haematology: Elsevier Churchill Livingstone; 2011. 650 p.” 

Results 

14. Reference values are supposed to be determined using apparently healthy individuals. Inclusion of the Hepatitis B positive individuals will affect the determined values. Please exclude these individuals from the calculations Data from hepatitis B positive individuals have been excluded and data re-analysed. Page 6; line 145

15. Presentation of the reference intervals in a tabular form will make it easier for the ready than the figures All the reference intervals have been presented in tabular form (see table 4) Page 14

16. The minimum of 120 individuals recommended by the CLSI for the determination of reference intervals is to make it possible to calculate 90% confidence intervals around the lower and upper values. The confidence intervals seem to be missing in the manuscript. Manuscript has been revised to include the 90% confidence intervals for both the lower and upper limits; see table 4. Page 14

Discussion, References 

17. The discussion is well written but could be updated with more references comparing similar studies. More references added 

18. the reference section looks too short. Kindly update. References added 

REVIEWER 2 

Abstract: 

1. Suggest to remove "Deranged" and replace with "Abnormal"

- Suggest to include weeks of gestation for each trimester, space allowing Done Page 2; line 29

Page 2; line 35 – 36 

Methods 

1. Exclusion criteria: Please clarify whether individuals with acute illness at the time of collection and/or use of prescribed medication (e.g. thyroid supplementation) Data from pregnant women with clinically diagnosed acute or chronic diseases were excluded from the study.

The only recorded medications the participants were on were the “routine pregnancy drugs” The inclusion criteria section has therefore been revised to include:

“All participants were on the following medications: folic acid (5 mg daily), oral iron tablets (200 mg three times daily) and oral multivitamin tables; these medications are generally called routine pregnancy drugs within the Ghana Health Services as they are part of the routine medical care for all pregnant women.” Page 6; line 138 – 141. 

Study participants: Please clarify whether iron supplementation was captured during assessment. 

In addition, was any longitudinal assessment evaluated at the end of the pregnancy? Iron supplementation is mandatory for all pregnant women in Ghana as part of routine care. This fact has been included in the 

No Page 6; line 138 – 141

- Reference Interval Determination: Please add confidence intervals for each estimate. CLSI guidelines recommends reporting 90% CIs

 Manuscript has been revised to include the 90% confidence intervals for both the lower and upper limits; see table 4. Page 14

Experimental Protocols: There is insufficient information in this section. For every assay, the manufacturer analyzers should be reported as well as their method (i.e. traceability) and performance throughout study duration (i.e. CV for QC levels). 

In addition, multiple centers were included in this study - were any center-specific differences in methodology evaluated? Each site uses 5-part Sysmex haematology analyser. Since this is a retrospective data that was retrieved from the hospital data management system, we were unable to collect the performance data of the analysers since the data was not prospectively collected. 

We believe this would have been valid undertaking if the design for the study was a prospective one since this would have allowed us to interrogate the comparability of the laboratory results as well as inter- and intra-laboratory agreement. 

- Data Analyses: CLSI guideline recommends using the Harris and Boyd method to test for partitions (i.e. gestational-age specific differences). Please clarify why MW and KW tests were selected. The recommendation from Harris and Boyd was employed by comparing the standard deviations of parameters per trimester (table 3). 

The MW and KW statistical procedures were only employed to test statistical differences between the haematological values. 

In addition, please clarify if outliers were removed and whether normality of data were assessed prior to statistical testing. The data was explored for outliers using the recommendation from the CLSI prior to analyses.

Data was assessed for normality before choosing non-parametric means was selected for the data selection. 

Results: 

Figure 1: Suggest to include these graphs for every parameter in supplemental methods. Noted 

Table 1: Please clarify if individuals were excluded on the basis of positive Hep B status. Data from hepatitis B positive individuals have been excluded from the revised manuscript; see under the revised “Exclusion criteria section”. Page 6; line 145.

Table 3: Suggest to include RIs in this table as well (or an additional table) rather than in the graphs. The rational for including table 3 is to provide basis for why a separate or combined reference intervals should be used. The reference intervals are included in table 4. Page 14

In addition, was the differential evaluated? No 

Discussion: 

The discussion mainly focuses on Hgb levels and their classification based on WHO standards. It should be highlighted that the WHO standards may not suit the ethnic population of Ghana and it is potentially not a public health problem. If no pregnant individuals or their neonates had serious complications, should this be considered abnormal? Suggest to include a comment on this. We have acknowledged the cross-sectional nature of the study as a key limitation as we could not access pregnancy outcome data. Page 19; line 374 – 386.

---

## [Decision Letter · Decision Letter 1]

1 Jun 2022

PONE-D-22-04279R1Establishing pregnancy-specific haematological reference intervals in Ghana; a three-center cross-sectional study.PLOS ONE

Dear Dr. Adu,

Thank you for submitting your manuscript to PLOS ONE. After careful consideration, we feel that it has merit but does not fully meet PLOS ONE’s publication criteria as it currently stands. Therefore, we invite you to submit a revised version of the manuscript that addresses the points raised during the review process.

We look forward to receiving your revised manuscript.

Kind regards,

Orvalho Augusto, MD, MPH

Academic Editor

PLOS ONE

Journal Requirements:

Reviewers' comments:

Reviewer's Responses to Questions

**Comments to the Author**

1. If the authors have adequately addressed your comments raised in a previous round of review and you feel that this manuscript is now acceptable for publication, you may indicate that here to bypass the “Comments to the Author” section, enter your conflict of interest statement in the “Confidential to Editor” section, and submit your "Accept" recommendation.

Reviewer #1: All comments have been addressed

Reviewer #2: (No Response)

2. Is the manuscript technically sound, and do the data support the conclusions?

Reviewer #1: Partly

Reviewer #2: Yes

3. Has the statistical analysis been performed appropriately and rigorously? 

Reviewer #1: Yes

Reviewer #2: Yes

4. Have the authors made all data underlying the findings in their manuscript fully available?

Reviewer #1: Yes

Reviewer #2: Yes

5. Is the manuscript presented in an intelligible fashion and written in standard English?

Reviewer #1: Yes

Reviewer #2: Yes

6. Review Comments to the Author

Reviewer #1: In Table 4 (page 14), the 90% confidence interval for the upper limits of the following parameters do not seem technically or clinically possible: HCT (351), MCHC (418 and 223), RDW (447). I would suggest to remove these columns.

Reviewer #2: Thank you for submitting this revised version of the manuscript. Most comments have been addressed, but select points still need to be considered.

- Both reviewers commented on the availability of quality control data for the hematology system in use. This needs to be included to ensure proper analytical performance was obtained throughout the study (i.e. was internal QC run?)

- In my opinion, graphs need to be provided by gestational age. Otherwise, it is very difficult to visualize the data and remark on observed trends. This would be very important to laboratories interpreting these findings.

- A reference was added for the >13 cut-off, but there is insufficient rationale to justify its use. Please provide more in-depth rationale and discuss the impact of high HGB on pregnancy outcomes

7. PLOS authors have the option to publish the peer review history of their article (what does this mean?). If published, this will include your full peer review and any attached files.

Reviewer #1: No

Reviewer #2: No

---

## [Author Response · Author response to Decision Letter 1]

15 Jun 2022

RESPONSES TO REVIEWERS’ COMMENTS

Establishing pregnancy-specific haematological reference intervals in Ghana; a three-center cross-sectional study (manuscript ID: PONE-D-22-04279)

COMMENTS RESPONSE PAGE

1. In Table 4 (page 14), the 90% confidence interval for the upper limits of the following parameters do not seem technically or clinically possible: HCT (351), MCHC (418 and 223), RDW (447). I would suggest to remove these columns. Table 4 has been revised to address this; a footnote “Note: the upper limit of the 90% CI of HCT, MCH, MCHC, and RDW for T2 are not indicated as they were very large to be considered technically feasible” has been added to highlight this change. Page 14, line 281 – 282 

2. A reference was added for the >13 cut-off, but there is insufficient rationale to justify its use. Please provide more in-depth rationale and discuss the impact of high HGB on pregnancy outcomes. Manuscript has been revised to include additional information “Maternal haemoglobin and adverse birth outcomes have been shown to follow a U-shaped curve where both low and high gestational haemoglobin are associated with adverse pregnancy outcomes such as preterm, gestational diabetes, stillbirth, pre-eclampsia, and small-for-gestational age (1-3). Therefore, in accordance with previous meta-analyses report (4), intra-pregnancy Hb >13.0 g/dL was classified as high maternal haemoglobin to classify participants with high gestational haemoglobin.” Relevant references have also been cited. Page 8, line 169 – 174 

3. Both reviewers commented on the availability of quality control data for the hematology system in use. This needs to be included to ensure proper analytical performance was obtained throughout the study (i.e. was internal QC run?) Per the prevailing protocols in each of the participating laboratory, control samples (low, normal and high) are run daily as quality control measure/check on analyser performance prior to assaying patient sample. Only when the respective haematology analyser passes the quality control tests is the analyser used to assay patient samples. Additionally, each of these daily quality control sample data (low, normal and high) are cumulatively evaluated monthly as a quality assurance measure to identify trends in analyser performance that may require immediate action. Page 7, line 161 - 167

4. - In my opinion, graphs need to be provided by gestational age. Otherwise, it is very difficult to visualize the data and remark on observed trends. This would be very important to laboratories interpreting these findings. This has been addressed in the revised manuscript; in figure 1A, [haemoglobin] is compared per trimester; figure 1C & 1F are specific trimester 2 [Hb] comparison per age category and participants’ place of residence; figure 1D & 1G are specific trimester 3 [Hb] comparison per age category and participants’ place of residence 

- 

1. Dewey KG, Oaks BM. U-shaped curve for risk associated with maternal hemoglobin, iron status, or iron supplementation. Am J Clin Nutr. 2017;106(Suppl 6):1694S-702S.

2. Abeysena C, Jayawardana P, de ASR. Maternal haemoglobin level at booking visit and its effect on adverse pregnancy outcome. Aust N Z J Obstet Gynaecol. 2010;50(5):423-7.

3. Díaz-López A, Ribot B, Basora J, Arija V. High and Low Haemoglobin Levels in Early Pregnancy Are Associated to a Higher Risk of Miscarriage: A Population-Based Cohort Study. Nutrients. 2021;13(5).

4. Young MF, Oaks BM, Tandon S, Martorell R, Dewey KG, Wendt AS. Maternal hemoglobin concentrations across pregnancy and maternal and child health: a systematic review and meta-analysis. Ann N Y Acad Sci. 2019;1450(1):47-68.

---

## [Decision Letter · Decision Letter 2]

27 Jun 2022

PONE-D-22-04279R2Establishing pregnancy-specific haematological reference intervals in Ghana; a three-center cross-sectional study.PLOS ONE

Dear Dr. Adu,

Thank you for submitting your manuscript to PLOS ONE. After careful consideration, we feel that it has merit but does not fully meet PLOS ONE’s publication criteria as it currently stands. Therefore, we invite you to submit a revised version of the manuscript that addresses the points raised during the review process.

We look forward to receiving your revised manuscript.

Kind regards,

Orvalho Augusto, MD, MPH

Academic Editor

PLOS ONE

Journal Requirements:

Additional Editor Comments:

This is a very important report and I feel sad to still find issues.

Please respond to the reviewer's question.

1. Line 228 - I think the "semester" should be "trimester"

2. For the prevalence of anaemia, please add 95% confidence interval (exact binomial confidence interval. In fact, this could be a separate figure or table.

3. The lines between 226 to 262 fail to describe the results properly. Rather than talking about the medians being compared per different groups, the authors focus only on p-values.

Reviewers' comments:

Reviewer's Responses to Questions

**Comments to the Author**

1. If the authors have adequately addressed your comments raised in a previous round of review and you feel that this manuscript is now acceptable for publication, you may indicate that here to bypass the “Comments to the Author” section, enter your conflict of interest statement in the “Confidential to Editor” section, and submit your "Accept" recommendation.

Reviewer #1: All comments have been addressed

Reviewer #2: All comments have been addressed

2. Is the manuscript technically sound, and do the data support the conclusions?

Reviewer #1: Yes

Reviewer #2: Yes

3. Has the statistical analysis been performed appropriately and rigorously? 

Reviewer #1: Yes

Reviewer #2: Yes

4. Have the authors made all data underlying the findings in their manuscript fully available?

Reviewer #1: Yes

Reviewer #2: Yes

5. Is the manuscript presented in an intelligible fashion and written in standard English?

Reviewer #1: Yes

Reviewer #2: Yes

6. Review Comments to the Author

Reviewer #1: Thank you for submitting a revised manuscript.

There are still very abnormal values indicated in Table 4 for the 90% CI for trimester 3 MCHC (223) and trimester 2 TWBC (87). As indicated in the earlier review comment, several of the upper values of the 90% CI are not clinically feasible and recommended the column could be removed.

Reviewer #2: (No Response)

7. PLOS authors have the option to publish the peer review history of their article (what does this mean?). If published, this will include your full peer review and any attached files.

Reviewer #1: No

Reviewer #2: No

---

## [Author Response · Author response to Decision Letter 2]

8 Aug 2022

Comment Responses Page 

Editor comment 

1. Line 228 - I think the "semester" should be "trimester" Correction effected Page 11; line 230.

2. For the prevalence of anaemia, please add 95% confidence interval (exact binomial confidence interval. In fact, this could be a separate figure or table. Manuscript revised to include the 95% CI (exact binomial confidence interval) “Furthermore, whereas 46.5% of participants had low haemoglobin level and therefore anaemic, 9.3% had higher haemoglobin levels. When participants haemoglobin levels were considered per trimester, 36.4% (95% CI: 0.1719 – 0.5934), 31.6% (95% CI: 0.2487 – 0.3904) and 68.2% (95% CI: 0.5951 – 0.7601) respectively of participants in trimester 1, 2 and 3 were anaemic.”

However, since the main import of the manuscript is that using the Caucasian-centric reference intervals (which was the basis for the anaemia prevalence in table 2), may lead to some participants incorrectly classified as anaemic, we are of the opinion that constructing additional table/figure will not be necessary in this instance. Page 10; line 216 - 220

3. The lines between 226 to 262 fail to describe the results properly. Rather than talking about the medians being compared per different groups, the authors focus only on p-values. The description to the figure has been revised to include the respective median to enable the reader better appreciate the data. The revised manuscript thus read “Participants’ haemoglobin levels were compared with the demographic data (figure 1). Overall, the median haemoglobin significantly differed across the trimesters of pregnancy. The median haemoglobin was higher in trimester 1 (11.80 g/dl) but decreased progressively across trimesters 2 (11.00 g/dl) and 3 (10.50 g/dl) [fig 1A: p = 0.0232 (trimester 1 vs trimester 3); p = 0.0005 (trimester 2 vs trimester 3)]. There was a general trend toward higher median haemoglobin levels with advancing age of participants as the respective median haemoglobin levels for 14 – 19 years, 20 – 29 years, 30 – 39 years, and 40 – 49 years were 10.35 g/dl, 10.80 g/dl, 11.00 g/dl, and 11.10 g/dl [fig 1B; p = 0.049 (14 – 19 years vs 30 – 39 years group; p = 0.0278 (14 – 19 years vs 40 – 49 years group)]; a similar trend was also observed when data was explored for trimester 2 (figure 1C) and trimester 3 (figure 1D) although the differences did not reach statistical significance. Also, the median haemoglobin level was significantly higher in participants from Ashanti region (11.80 g/dl) compared to median haemoglobin level of participants from Eastern region (10.50 g/dL) or Upper Est region (10.30 g/dl) [figure 1E: p <0.0001 (Ashanti vs Eastern region); p <0.0001 (Ashanti vs Upper West region)]. Similarly, the median haemoglobin levels in trimester 2 was significantly higher in participants from Ashanti region (11.80 g/dl) compared to participants from Eastern (10.80 g/dl) or Upper West (10.30 g/dl) regions (figure 1F: p = 0.0031 vs Eastern region; p <0.0001 vs Upper West region). Moreover, the median haemoglobin in trimester 3 was higher in participants from Ashanti region (11.65 g/dl) compared to the Upper West (10.50 g/dl) region (figure 1G: p = 0.0240).” Page 12; line 227 – 246.

Reviewer 1 

There are still very abnormal values indicated in Table 4 for the 90% CI for trimester 3 MCHC (223) and trimester 2 TWBC (87). As indicated in the earlier review comment, several of the upper values of the 90% CI are not clinically feasible and recommended the column could be removed. Corrections effected Page 15; line 284 - 289

---

## [Decision Letter · Decision Letter 3]

30 Aug 2022

Establishing pregnancy-specific haematological reference intervals in Ghana; a three-center cross-sectional study.

PONE-D-22-04279R3

Dear Dr. Adu,

We’re pleased to inform you that your manuscript has been judged scientifically suitable for publication and will be formally accepted for publication once it meets all outstanding technical requirements.

Kind regards,

Orvalho Augusto, MD, MPH

Academic Editor

PLOS ONE

Additional Editor Comments (optional):

This is very important work to help establish appropriate clinical guidelines for African pregnant women.

I still cannot understand why the authors still test for normality when in the end used non parametric methods.

Reviewers' comments:

Reviewer's Responses to Questions

**Comments to the Author**

1. If the authors have adequately addressed your comments raised in a previous round of review and you feel that this manuscript is now acceptable for publication, you may indicate that here to bypass the “Comments to the Author” section, enter your conflict of interest statement in the “Confidential to Editor” section, and submit your "Accept" recommendation.

Reviewer #1: All comments have been addressed

Reviewer #2: All comments have been addressed

2. Is the manuscript technically sound, and do the data support the conclusions?

Reviewer #1: Yes

Reviewer #2: Yes

3. Has the statistical analysis been performed appropriately and rigorously? 

Reviewer #1: Yes

Reviewer #2: Yes

4. Have the authors made all data underlying the findings in their manuscript fully available?

Reviewer #1: Yes

Reviewer #2: Yes

5. Is the manuscript presented in an intelligible fashion and written in standard English?

Reviewer #1: Yes

Reviewer #2: Yes

6. Review Comments to the Author

Reviewer #1: The earlier recommendation to remove all the 90% CIs has not rather been replaced with "-" for the abnormally high limits. Keeping some values and putting "-" makes it appear the limit is to infinity.

Reviewer #2: The author has addressed all remaining comments by reviewers. Updates to Table 4 and results section are satisfactory.

7. PLOS authors have the option to publish the peer review history of their article (what does this mean?). If published, this will include your full peer review and any attached files.

Reviewer #1: No

Reviewer #2: No

---

## [Editor Report · Acceptance letter]

5 Sep 2022

PONE-D-22-04279R3 

Establishing pregnancy-specific haematological reference intervals in Ghana; a three-center cross-sectional study. 

Dear Dr. Adu:

I'm pleased to inform you that your manuscript has been deemed suitable for publication in PLOS ONE. Congratulations! Your manuscript is now with our production department. 

Kind regards, 

on behalf of

Dr. Orvalho Augusto 

Academic Editor

PLOS ONE